# An Agile Digital Platform to Support Population Health—A Case Study of a Digital Platform to Support Patients with Delirium Using IoT, NLP, and AI

**DOI:** 10.3390/ijerph18115686

**Published:** 2021-05-26

**Authors:** Mohan R. Tanniru, Nimit Agarwal, Amanda Sokan, Salim Hariri

**Affiliations:** 1Mel and Enid Zuckerman College of Public Health, University of Arizona, Phoenix, AZ 85004, USA; aesokan@arizona.edu; 2Department of Internal Medicine, Banner University Medical Center, University of Arizona, Phoenix, AZ 85004, USA; NimitAgarwal@arizona.edu or; 3College of Electrical and Computer Engineering, University of Arizona, Tucson, AZ 85721, USA; hariri@arizona.edu

**Keywords:** population health, value cycles, digital health, dynamic software ecosystems, delirium patient care, agile digital platform

## Abstract

For an organization to be customer centric and service oriented requires that it use each encounter with a customer to create value, leverage advanced technologies to design digital services to fulfill the value, and assess perceived value-in-use to continue to revise the value as customer expectations evolve. The adaptation of value cycles to address the rapid changes in customer expectations requires agile digital platforms with dynamic software ecosystems interacting with multiple actors. For public health agencies focused on population health, these agile digital platforms should provide tailored care to address the distinct needs of select population groups. Using prior research on aging and dynamic software ecosystems, this paper develops a template for the design of an agile digital platform to support value cycle activities among clinical and non-clinical actors, including population groups. It illustrates the design of an agile digital platform to support clients that suffer from delirium, using digital services that leverage Internet of Things, natural language processing, and AI that uses real-time data for learning and care adaption. We conclude the paper with directions for future research.

## 1. Introduction

Businesses have recognized the importance of understanding the customer journey and reacting quickly to address customers’ changing expectations in an evolving technology landscape [1]. Such a customer-centric and service orientation means businesses have to use each encounter with a customer or service exchange to create value propositions and assess their perceived value-in-use or value-in-context to revise these value propositions as the customer expectations change [2]. By using such a value lens to create, fulfill, and assess value-in-use, often using digital services (i.e., services that create value supported by advanced technologies), businesses can create differentiated value in a competitive marketplace. More importantly, such a market dynamic calls on businesses to use agile organizational models to innovate and explore new value propositions [3]. Depending on the customer value cycle (time between two purchases of the same product/service) and product life cycle (time to create a new product/service to fulfill value), this also means that businesses may need to use multiple value cycles to keep customers engaged and empowered [4].

This market dynamic is blurring the difference between business and IT strategies. In fact, it is interweaving or fusing these two strategies under the framework of the digital business strategy [5], depending on where value is created (e.g., productivity within the business or value creation to support customer needs). When value creation is driven by the changes in customers’ needs, the digital business strategy has to develop innovative digital services that create, fulfill, and assess value-in-use. Analyzing feedback to improve clinical diagnosis (i.e., clinical informatics), business performance (i.e., business analytics), or system performance (i.e., analytics in general) is necessary when there are changes in the inputs, or models used to process these inputs. When such changes are driven by evolving needs of the external consumer, feedback to improve value created by an organization comes from a consumer’s perception of such value (value-in-use).

Independent of where the feedback comes from, a software ecosystem designed to improve performance has to be agile to support the faster design of these new digital services, or the reconfiguration and/or redesign of the current digital services to meet evolving customer expectations. Just as businesses are leveraging multiple partners to fulfill value, today’s software ecosystems also have to leverage the expertise and resources of multiple software developers and technology partners as they design digital services to create and fulfill value [6]. In fact, the digital platforms used to support the communication of information and the coordination of value cycle activities across business and customer ecosystems need an agile software ecosystem that leverages a mix of technologies and application environments (e.g., mobile apps, web browsers, IoT sensors, cloud base services, etc.) to create and sustain value [7].

Designing an agile digital platform to address customer needs is an even bigger challenge for healthcare organizations such as hospitals. In a business ecosystem, value cycles that leverage external partners are often under the control and coordination of the business. The business can therefore direct value creation and value fulfillment, and it has some influence on gathering feedback on value-in-use. However, health systems, even if they are inherently service driven, must operate two different value cycles. The first value cycle is within a provider ecosystem (e.g., hospital) where the initial value is created (e.g., a health condition diagnosis) and some value is fulfilled (e.g., testing of patients and treatment plan design when they are in-patients). The second and often much longer value cycle is within the patient ecosystem (e.g., a patient’s home or at a temporary facility such as a skilled nursing or rehab facility). The value fulfillment, value-in-use tracking for feedback, and even new value creation (e.g., changes to treatment plans) are supported by multiple external clinical and non-clinical partners, including patients and health care providers. While digital platforms used to support the first value cycle can be designed for agility with healthcare provider oversight, even when they use several external software partners [8], building an agile digital platform for the second value cycle is much more complex.

With increased access to technologies such as wearables and mobile apps to monitor health conditions and track physical activities, and with the growing use of telehealth, portals and social media tools to seek and share information, the patient demand for digital services to create and fulfill value in healthcare is growing. At the same time, healthcare providers are looking to use patient interest in self-managing their care at home to shift a significant part of the value fulfillment and successive value cycles into the patient ecosystem to reduce readmission costs and improve patient satisfaction. For public health agencies with a focus on prevention, most of the value cycle activities to tailor practices to create value, fulfill value, and assess value-in-use have to be done within the population or patient ecosystem using a number of external actors, both clinical and non-clinical. Hence, the following research question: How can we build an agile digital platform to support health care delivery in a client ecosystem? The client here refers to both the patients seeking care transition services and the population groups looking to adhere to preventive healthcare practices.

We will try to answer this question in the paper using the following three steps: (1) design innovative care delivery models that create value for a specific client population; (2) develop an agile digital platform that uses configurable digital services to communicate information and coordinate activities in support of value cycle activities; and (3) illustrate how such an agile digital platform using artificial intelligence (AI) and Internet of Things (IoT) technologies can be developed to support a particular client group, being patients who suffer with delirium conditions in this case. The paper is organized as follows to discuss each of these three steps:

Section 2 focuses on the first step. With a growing proportion of an aging population needing to address multiple chronic care conditions, and with healthcare systems looking to design care delivery models to reduce costs by shifting chronic care management to clients’ homes using technologies, the challenge becomes one of tailoring care to such a population effectively. The individuals in this age bracket have varying social, technological, and educational capabilities to self-manage their care at home, and they need a mix of clinical and non-clinical care providers to support their care. Using research on aging, this section develops a set of services that can be used to create value that is tailored to the specific client’s health condition and ecosystem needs;

Section 3 focuses on the second step. For the services identified in Section 2, we will discuss the strategies used to design a digital platform to support value cycle activities. With technologies helping clients monitor their health conditions and communicate with healthcare providers, connecting the clients with various clinical actors (labs, pharmacies, etc.) for treatment adherence, and non-clinical actors for social and emotional support, the digital platform must work with configurable digital services that evolve as the client needs and actors supporting these needs evolve. Using research on agile digital platforms, this section develops a template for the design of an agile digital platform to support value cycle activities;

Section 4 focuses on the third step. It illustrates the design of an agile digital platform to support clients that suffer from delirium. With the growing use of the IoT to monitor a client’s health condition in real time, and with the availability of AI technologies to analyze a large volume of real-time data for learning and adaption [9], the digital platform will address the client needs in a hospital. While its use is discussed within a provider context, it can be used in a home care environment as well with adaptations;

Section 5 provides some concluding comments and Section 6 provides some directions for future research.

## 2. Research on Aging and Services to Create Value

The term “aging” generally refers to our understanding of how we as human beings progressively deteriorate in our health during our adult period of life [10]. When measured on the life calendar, it often becomes a discrete number, and societies worldwide use such a number to make assumptions about people’s ability to contribute to economic productivity (e.g., setting an age for retirement), determine eligibility for government assistance (e.g., health insurance), and use statistical models to classify disease conditions such as chronic care conditions (e.g., how a certain percentage of the older population contributes to healthcare costs). Even health systems design care delivery models to provide care at home, with the assumption that the digital services needed to deliver care have to support a population with lower levels of visual and mobility capabilities, technological literacy, and healthcare knowledge. This often induces bias in the way the systems are designed and services are identified to create value. Therefore, stereotyping all adults over a certain age as a single monolithic group for designing health services to deliver care at home leads to missed opportunities [11].

Aging is a process, and innovations in medical science for clinical diagnosis and drug-based therapies have been extending human life, thus transforming the older population into a mix of sub-population groups with varying capabilities to manage their health conditions. The concept of active aging, defined by the World Health Organization [12], uses eight classes of determinants to gather information for the possible clustering of services for such populations. These include the physical environment, access to health and social services, individual behaviors (healthy behaviors, addictions, medications), personal characteristics (biology, genetics, psychology), support ecosystem (social support, violence and abuse, education and literacy), economic factors (income, social protection, work), gender, and culture. Overlying these determinants are other factors often considered in social diagnosis [13], such as the roles people play (within their ecosystem), the relationships they have (with others in their network), their reactions to ecosystem conditions (fear, stigma, stress, etc.), and the resources they have (physical, economic, emotional). All of these can influence the clients’ abilities to manage their health conditions as some of these factors change. This means that healthcare value delivery is a dynamic process, and the services used to create value must continue to evolve based on the feedback from value-in-use.

For the purposes of this discussion, we will use prior research to classify services that create value into the following four types: self-health management, civic engagement, caregiver engagement, and community engagement. While certain older population groups today have limited literacy on health and technical knowledge, more baby boomers are entering this age bracket and are increasingly conversant with and interested in using technology to self-manage their health conditions. With a greater capacity to contribute, these older populations can engage in many civic activities that benefit both the community and the health and wellbeing of the older adults. As health conditions deteriorate, some begin to rely on services provided by professional or informal caregivers, such as family members. Lastly, some older populations continue to face health inequities by virtue of economic, social, and geographical disparities, and need the support of social services to overcome barriers for access to quality care. We will look at each in detail below.

### 2.1. Self-Health Management Services

Several digital services to support population engagement at home are designed to allow people to self-manage their health conditions. This includes sharing health information, asking specific questions, seeking consultation, scheduling appointments, etc. The technologies used to support these services can include patient portals, websites, and digital health infomediaries [14]. The recent health emergency (COVID-19), in fact, has made many of these digital services important for care delivery at home (e.g., telehealth technologies). The effectiveness of such digital services depends on how engaged both the patient and health care providers are in frequent communication and follow-ups in the area of mental health [15]. Remote monitoring services are used, from tracking blood sugar levels to other complex chronic care [16]. Social media technologies, such as digital infomediaries, are used for consultation and seeking answers to medical questions [17].

In summary, there are important opportunities to create value for older clients who want to self-manage their health conditions.

### 2.2. Civic Engagement Services

Even with medical advances and economic prosperity increasing longevity and well-being, and moving more people into the broad category of the older population, ageism continues to stigmatize this population in the labor market [18] and marginalize their role in productive social engagement [19]. Active aging strategies include promoting physical activity through volunteering and socializing [20]. These strategies can allow cities to use multiple agencies to make public places health friendly [21,22] and engage the older population in community-based voluntary activities [23]. Such an engagement in social, economic, cultural, spiritual, and civic affairs can help reduce isolation and keep the older population both physically and mentally active [24]. It can also lead to lower risks of morbidities, disability, and cognitive decline, and their civic engagement can lead to better physical and mental health, higher cognitive function, decreased loneliness, etc. [25]. In summary, the engagement of the older population in voluntary services (e.g., social care, and recreational and local community work) not only benefits society, but also improves the social and emotional health of the older population [26,27,28].

In summary, such a holistic and life-course approach means that the services used to create value need to use a temporal lens, as the shorter and longer term needs of the older population change over time. While health systems cannot be responsible for providing a broader range of services and varying them over time, they need to leverage a network of actors, including private and government sectors and social entrepreneurs that focus on providing social benefits [29,30,31].

### 2.3. Caregiver Engagement Services

Even as the demand for home-based care for older adults is rising across many high-income countries [32], the nature of this care is becoming multifaceted. The care is sometimes designed to support both older adults and their informal caregivers, especially when adults living at home start to lose mental capacities and the ability to care for themselves. In other words, digital services have to meet the needs of both the patient client and the caregiver client. The client services have to be time sensitive as the health conditions continue to deteriorate by using services that test their physical and cognitive capabilities to create value, such as reducing client falls or engaging in activities that improve cognition [33,34]. The role of caregivers, informal as well as specialized, is growing, and the challenge is to create cohesion between the services provided by the healthcare provider and other actors while addressing risks [35,36,37]. Providing reliable information and supporting trust building opportunities are key to engender the trust of clients in seeking such services.

In summary, caregivers are unpaid family members, and they provide an important means to reach and support this patient client population [38,39]. The services provided should include services that will help caregivers manage their own health while caring for others. Such caregiver services are even more important when the caregiver lives with the person they are caring for [33]. With the availability of informal caregivers expected to decrease as many families are not engaging in co-residency, the communication and coordination of the caregiver and patient services requires a mix of value fulfillment activities from multiple actors [40,41,42].

### 2.4. Community Engagement Services

While medical advances are increasing the life span, they are not necessarily influencing the economic status of many older adult citizens living under or near poverty [43,44]. Many hospitals are using community providers to address the social determinants that contribute to health inequities (e.g., supporting transportation, nutrition assistance, childcare, or education). Community health workers, peer navigators, etc., are often used to assist clients with mental health problems, sexually transmitted diseases, and diabetes [45,46]. Some social agencies leverage the support of other social agencies, particularly when delivering care-related services is complex, as in supporting homeless populations [47]. With many social factors determining a client’s ability to manage their health condition, the challenge is to look upstream for prevention and downstream for care transition post-hospital discharge in order to identify the services needed and actors to engage with [48].

In summary, older adults with health inequities have limited resources and technologies, such as limited wireless access and/or smart phones. As seen during the COVID-19 health emergency, these clients are at a greater risk for infection and mortality [49], and their self-isolation for safety led to psychological and social challenges [50,51]. Therefore, innovative ways to deliver care to such populations has become critical during the COVID-19 lockdowns [52], including the use of a mix of digital and non-digital services (e.g., telephone, local community workers, etc.) to maintain social connections considered essential [53] despite the technology barriers [54,55].

While this classification is not exhaustive, it does provide a range of services to older clients that span from self-health management to continual health monitoring in both supportive and less than supportive ecosystems [46]. Using these four different service types, Figure 1 below identifies a list of services needed to support these clients. As discussed earlier, the goal here is not to build a point-to-point solution that supports one group by one provider and its partners, but to develop a platform that is agile to address client needs as they evolve or change. The value fulfillment services that will become a part of a digital platform are discussed in the next section.

## 3. Designing a Digital Platform to Support Agility

A client-centric and value-driven service model to address care delivery outside a health system is a different paradigm for healthcare systems [56]. It represents a shift from the provider-centric view of “the doctor knows best” to a “patient-centric” view where care delivered at home is best for the patient and the provider. Healthcare organizations that have longer value cycles and multiple actors outside their ecosystem fulfilling many value cycle activities, must transform their care delivery models using advances in technologies. This is especially critical as chronic care conditions continue to predominate in older populations.

In the drive to use technologies, one cannot ignore the human element when it comes to digital service use. For example, digital services that leverage sensors in wearables to drive behavioral change such as reducing obesity, will not be effective if the clients do not know how to use these digital services. Web-based interventions or remote monitoring services designed to measure hemoglobin A1c or blood pressure have limited impact on client behavior without adequate training [57,58,59]. When digital services are supported by social and community health workers that provide education and answer questions, it improves their use [46,60]. When patient portals used to disseminate information are contextualized to lead clients to ask questions and seek clarification, they can prove to be of greater value [61]. Even with intuitive interfaces, such as videos to support physical or mental exercises and calming music to reduce pain, digital services will have a limited impact when the patients have mobility challenges. Alternative technologies, such as virtual reality (VR) goggles with animated games or travel to distant places, are shown to be effective in reducing clients’ pain [62,63].

The examples discussed above highlight the importance of understanding the client context or ecosystem where the value is perceived, so that care delivery models can be effective. However, context is not static and client capabilities and expectations continue to change. As clients become more experienced in using digital services and build their capacity to self-manage their health condition by asking questions, seeking clarification and second opinions, etc., the digital services have to adapt to address the changes in client capabilities and expectations. In addition, the social ecosystem around a client may change as well, such as their roles, relationships, reactions to their health condition, and resources [13]. This makes client context dynamic, and continual value-in-use feedback is needed to adapt the digital services to this evolving context. Implicitly, the mental model or the “lens”, within which a client interprets the value created, evolves over time [64]. This is especially the case when a client condition deteriorates and the role of others (e.g., caregivers) increases, or when a client moves from a supportive ecosystem to a less than supportive ecosystem. This has been seen during the COVID-19 pandemic, with many clients moving from preventive care services to address food and economic insecurities.

Without claiming exhaustiveness, we will discuss the following three ways to make digital services adapt to a changing client context: increase value cycle frequency, broaden the scope of value-in-use assessment, and broaden the scale of value-in-use assessment.

### 3.1. Increase Value Cycle Frequency

The goal here is to make digital services adaptive by gathering feedback frequently to track changing client expectations. The increased frequency of information gathering has been shown to improve behavioral changes among clients. For example, the real-time monitoring of glucose levels with education has been shown to improve individual adherence [46], frequent feedback on sleep quality has shown improvement for clients with apnea [65], and research is ongoing on how adaptive feedback control systems with real-time feedback can improve smoking cessation behavior [66].

Remote and continual monitoring of clients generates large volumes of data, and tools such as AI have become useful in identifying changing customer needs and redesigning digital services accordingly. For example, an AI-assisted alert system was shown to be effective in monitoring patients in an intensive care unit or emergency rooms [67]. Data from a patient’s vital signs are analyzed to modify early warning scores that can lead to predicting a cardiac arrest [9]. Even routinely available data analyzed by providers can lead to decisions on how to alter external actor engagement in supporting clients (e.g., emergency management technicians visiting clients at home, specialists consulting nursing home patients, and staff changes in patient rooms) [46,68,69]. In summary, frequent feedback from clients during value-in-use may lead to not only a reconfiguration of digital services, but also broaden the scope of actors used in value-in-use assessment. This is discussed next.

### 3.2. Broaden the Scope of Value-in-Use Assessment

All innovative value propositions used to create value need the engagement of all the actors, including clients, in value fulfillment. When these value propositions change to meet customer expectations, it can lead to changes in the actors involved. The strategies used to support the adoption and diffusion of an innovation (i.e., value-in-use of a digital service) focus on client and digital service complexity [70]. In other words, value cycles have a predictable frequency (i.e., how quickly feedback is used to redesign digital services to create new value). However, changing the value cycle frequency to quickly monitor client behavior means altering the value fulfillment frequency, and this may alter the behavior of other actors. For example, consider a nurse calling system designed to use multiple buttons for the following different types of services: bathroom visits, pain management, and general information. When the speed to respond to a nurse call is altered to improve patient satisfaction, using text alerts and escalation protocols that transfer calls to others when response time is slow, this alters the value fulfillment activities of the nursing staff. If there is a difference in the perceived value-in-use of the patients (e.g., immediate response to a call made to nursing staff) and the perceived value-in-use of the nursing staff (e.g., staff’s perception of the “urgency of the call” depending of the call source, for example surgical or cardiac patients vs. oncology or neurology patients) when the frequency of the value cycle is changed, it can lead to not meeting patient expectations [69,71]. In other words, increasing the value cycle frequency to address the context changes in the clients may have a negative impact, or be misaligned with the actors who are fulfilling the value created.

Such misalignment can be significant when some value fulfillment activities use machine actors (e.g., systems that send alerts when calls are made, escalate to others, etc.) and others use human actors such as nursing staff, whose activities cannot change quickly (e.g., process changes, policy implications, etc., take time to change). This is often reflected in model or algorithmic bias or fairness when these biases get embedded in digital services [72]. For example, the algorithms used to make predictions on heart conditions may be biased if they undercount women and especially women of color, impacting diagnoses and treatment plans [73]. When machine learning algorithms are used to make predictions based on large volumes of data, they can minimize the influence of a few outliers, leading to the misapplication of forecasted outcomes [74,75]. Research on the design for diffusion [76], and being aware of the situation as a whole [77], calls for viewing the client context more broadly as value fulfillment and value-in-use is occurring within a complex ecosystem with many actors besides the clients. Therefore, when the value cycle frequency is increased using machine actors to address the changes in context from client feedback, feedback is also needed from other actors influencing value fulfillment. Such broadening of the scope of value assessment (actors providing the feedback) means potentially broadening the scale of value assessment as well, which is discussed next.

### 3.3. Broaden the Scale of Value-in-Use Assessment

Technology-induced change has been studied in decision support systems [78], and clients using digital services to self-manage their care condition may seek different types of information not supported by any of the digital service modules or components. For example, as older peoples’ cognitive functions change over time, digital services used to support caregivers must evolve as the patients deteriorate in their health conditions. Even with limited access to mobile technology to care for their health conditions [79], vulnerable populations saw their service needs dramatically change during COVID-19 (from prevention to address food and economic insecurities). Those who used apps to manage their glucose levels have started to want to use the apps to apply for health insurance [46]. Given that many social determinants contribute to health inequities, addressing some of them using digital services may lead to clients seeking other services, thus implicitly calling for broadening the sources used to provide feedback, i.e., the scale of value assessment. As one uses a digital service (e.g., mobile health unit (MHU) calling or texting each client to provide information on COVID-19), it may lead to other services for which assistance is needed (food, transportation, etc.). This can lead to gathering feedback on these to create new services and altering current services given their interdependency.

In summary, digital services need to increase the frequency of value cycles to show less bias in fulfilling value and greater agility to meet evolving client expectations. They also need to assess any change in the scope (or depth) of all those involved in fulfilling the value, and change in the scale (or breadth) of the services needed to address the clients’ value expectations. The design of a digital platform to support such agility is discussed next.

### 3.4. Building an Agile Digital Platform

As organizations use agile models of structure and governance to support the exploration and evaluation of new services to create and fulfill value and leverage technologies to design digital services, they similarly need an agile software ecosystem that uses a number of digital service modules or components that can be easily configured to adapt to the evolving client needs. Figure 2 shows some of the service modules designed to support the select senior population segments discussed in the previous section. The software modules interact with each other, potentially across partner and client ecosystems, using a distributed digital platform. This platform supports the communication of information and the coordination of value cycle activities among all the actors involved. Each service module supports a distinct value proposition (e.g., searching for an external caregiver, tracking health conditions using a glucose monitor, finding a physician to get a second opinion). They can be configured to work together to create value for a client, such as a caregiver servicing a dementia patient or an older client looking to monitor their blood pressure.

A recent institute of electrical and electronics engineers (IEEE) special Issue on managing software ecosystems has recognized the need for agile ecosystems to support complex and dynamic environments such as the one discussed here [80]. There can be a mix of autonomous and automated systems tracking rapid changes in the external environment, and a mix of machine actors (software systems). Human actors (in the clinical and non-clinical ecosystem), including technology developers, need a digital platform to support their collaboration [81] and interactions to share data and collaborate their activities in real time [82]. Such a dynamic orchestration of digital services has shown success in multi-tenant business networks during run-time as actors join and leave [83], and among innovation ecosystems where ideas are generated and evaluated [84].

A dynamic software ecosystem for rapidly evolving client expectations, as shown in Figure 2, needs not only increased frequency of value assessment for reconfiguration, but an ability to address potential changes in the scope and scale of such an assessment. For example, if the run-time feedback on defined parameters (e.g., pain medication requests from a nurse call system or smart bed alerts for bathroom support) is lower than expected, what conclusions follow? Did it address a value proposition (i.e., patient satisfaction attributed to staff response)? Should it lead to broadening of the scale by seeking feedback on other services (e.g., food catering services, discharge services, etc.)? Should it lead to a deepening of the scope by seeking feedback from other actors in the network (e.g., nursing staff on changing patient conditions, improved care processes such as hourly rounding, etc.)? Such an insight on the scope and scale needed in value assessment requires an understanding of the client or actor intent and should be explicitly considered in the design of the digital services. 

In summary, gathering client intent if other value-in-use feedback is not sufficient to autonomously reconfigure digital services is needed if agile digital platforms and software ecosystems are to support a health system that is seeking to support client needs quickly using value cycle activities. One pilot application that combines client intent and a dynamic software ecosystem is discussed in the next section.

## 4. Digital Platform to Support Patients with Delirium

A fear of falling is often a contributor to fall injuries inside and outside hospitals [85]. Such a fear is a pervasive psychological problem in older people and not only contributes to falls [86], but also makes them avoid activities that can keep them healthy, i.e., less dependent on society and medications [87,88,89]. We will focus on one particular symptom of this, delirium, which contributes to the fear of falling. The digital platform that uses advanced technologies and digital services to create, fulfill, and assess value-in-use to support delirium patients is summarized in Figure 3, and is elaborated on in this section.

### 4.1. Clients

Delirium is considered a syndrome due to its multicomplex nature and is dangerousness. It is common in older hospitalized adults, with a prevalence of about 30–50%, and affects around seven million patients annually. A big challenge for healthcare systems is that this syndrome can go undiagnosed by healthcare team members, including nurses and physicians. This leads to a longer length of hospital stay, loss of physical function, which can lead to institutionalization at long-term care facilities, progression to dementia, and even death. The major challenges of the effective management of delirium include the need for a multidisciplinary approach, difficulties with risk modification, and a lack of effective low-risk pharmacological treatments. Thus, a system that can assist in early identification, by means of timely screening and assessment, is needed so that precipitating factors can be identified and removed for improvement in patient care. Clients with cognitive impairment can fall during “unexpected care windows”, and no amount of nursing resources can prevent these falls. However, such falls contribute to hospital admissions, medical treatments, and high insurance costs, not to mention the personal physical and psychological pain to the patients.

### 4.2. Technologies

Remote monitoring of such patients can be done using wearable-based, non-wearable-based, and fusion-based systems [90]. The wearable-based system uses sensors tied to various body parts, but are relatively inflexible and uncomfortable. The non-wearable-based systems, such as smartphone-based solutions, could be a very competitive alternative to the conventional wearable systems for fall detection [91]. However, as discussed earlier, when the value cycle frequency increases due to remote monitoring, multiple sources of information are needed to be analyzed, and fusion-based systems can help leverage diverse information sources. In the proposed system, information from visual clues are combined with audio responses to questions posed by the system to understand patient intent in creating and assessing value-in-use for patients.

In the digital platform proposed, called SeVa, the frontend used to create value is a mobile application written in the Apple iOS native program language Swift on Xcode 11, which runs on the iOS device with iOS 13. The chatbot engine is Dialogflow, a UI-based platform for creating smart and proactive chatbots. SeVA is a fusion-based system, and supports remote diagnosis and medical consultation. The SeVA backend server to fulfill value has four dedicated advanced reduced instruction set computer machines (ARMs) with 2GB memory and the Ubuntu Xenial system.

The remote monitoring service here includes a fusion of clues from the audio and visual content. It connects the following two different environments: SeVA Patient Room app (SPR), which is an application that resides in a patient room or ecosystem, and SeVA Master Control app (SMC), which is a master control application that resides in the provider ecosystem, connected to both the autonomous part of the digital service as well as the human actor engaged component, i.e., nursing station to react to specific patient context.

### 4.3. Value Creating Services

The value creating services are supported using a remote diagnosis that does not use text but rather visual gestures, such as the waving of hands or visual eye movement, as well as voice conversation. Both of these are important for elders or patients suffering with delirium. It uses personalized conversation to answer questions and provide advice. Specifically, the system creates value to clients by responding to patient gestures with consultation sessions. In addition, it does regular checks to ensure that the patient condition is monitored periodically, including a morning check and hourly rounding. It uses delirium checks to assess a patient’s cognitive capabilities and provides music to create a relaxed environment to reduce stress and/or help clients manage pain. Table 1 lists these patient services.

### 4.4. Value Fulfillment Service Modules

The service begins when the chatbot engine receives a trigger upon patient touch or sound. It sends the first sentence to the SPR and then waits for the patient’s response (i.e., returning sentence). The patient’s response will then be classified as the intent “Yes” or “No”, each leading to a different set of conversations. For periodic monitoring, such as hourly and morning checks, the SPR will ask a question every hour (except for the rest hour) to make sure the client’s needs are satisfied.

For delirium checks, the SPR starts by checking the patient’s delirium status by asking questions, or launches the delirium check game and uses cognitive assessment tests [92]. Two delirium check games are used at present. The first game, “Connect Node”, is used for testing visuospatial and executive ability. The patient is required to connect the node in a given order. If a patient fails the test, a message will be immediately sent to the SMC. The second game, “Click Animal”, tests patient attention. Pictures are displayed and disappear. The patient has to click the animal picture. The checking result will also be sent to the SeVA Master Control application. For supporting relaxation services, the SPR has embedded music created by the therapist in order to create a calm atmosphere and relieve panic.

Figure 4 shows the coordination of the value fulfilling activities using a particular scenario, beginning with an action from the patient. The fulfillment of the activities supported by the chatbot engine scenario are defined by medical staff.

A patient begins the scenario with an event, for example “waving hand.” When this patient request is received, the peripheral sensor recognizes the action and sends an hypertext transfer protocol secure (HTTPS) request to the SeVA backend server;The request is parsed to fetch the necessary information, such as the room number, from the database in order to trigger the chatbot engine to begin the conversation. The SeVA patient room application has a WebSocket connection with the chatbot engine so that the conversation can be executed;The conversation result is sent back to the SeVA backend server in the form of a SeVA command, and it is sent to the SeVA Master Control application in the nurse room;This leads to the nurse taking timely intervention if this is a critical event;The running status of the system is monitored continually by the monitoring server to ensure system reliability.

### 4.5. Agile Digital Platform to Support Activity Coordination among Actors

Since many of the components of the system are autonomous agents (or machine actors), the activity of each actor is discussed below. Some of these are running continually, and others are trigged based on new events. Within the actor-network theory, this human–machine–other human (e.g., nurse) interaction has to occur throughout the entire value cycle and let the value-in-use feedback be used to support agility in the software ecosystem. To coordinate the activities, the system here functions as discussed below.

Upon user registration and authentication, the user conversation can be initiated by hand waving, using the wake-up word, or simply touching the screen. Currently, a smart wristband detects the user movement and another sensor detects a patient’s wave gesture. Once an event is triggered, it will be routed to the chatbot engine to initiate the conversation using the SeVA mobile application.

The chatbot engine transforms the user’s voice into plain text, which is processed by the natural language processing (NLP) module. Using discrete word pre-processing and tokenizing, it is sent to a recurrent neural network (RNN) module. The output of the RNN is matching the word candidates if the patient poses a directly interpretable request or a suggested intent score of the patient’s question. If it is an intent score that needs further interpretation, it is then sent to the dialogue engine of the expert system module to decide on the final interpretation of the patient request. See Figure 5.

The medical expert defines skills, such as interpreting the intent and trigger, and replies. Once the intent matches with the skills, the system can begin the conversation autonomously. The skill used to begin the next conversation is derived from a knowledge base of skills, and this can be triggered either by a directly interpreted patient request or extracted from the interpretation using both the RNN and expert knowledge.

### 4.6. Agile Digital Platform to Support Information Sharing among Actors

The following five-layer framework is used to support the communication of data among the actors in the value cycle: infrastructure layer, data layer, scenario layer, application layer, and security layer. For more information on the infrastructure, see Appendix A.

The infrastructure layer provides the basic hardware requirement of the data collection unit. It includes the minimum requirement for deploying our system to a different environment, as most of our system service is on the cloud.

The data layer processes the incoming raw data from the infrastructure layer. The main tasks here are the text-to-speech conversion, speech-to-text conversion, and action recognition, using a neural network as the backbone. The patient action is recognized based on the temporal movement data, and a long-short-term memory neural network processes this input. The output of this layer will be the conversation plain text and the user action classified as a result.

The scenario layer contains the scenario provided to the professional medical expert and its return as a conversation presented to the user. More specifically, the medical expert predefines the scenario representation with the related incoming conversation plain text or action class. This will lead to a conversation and initiates a trigger to instantiate the next program or module in the application layer.

The application layer consists of the following two mobile applications: the SPR application and the SMC application. The SPR application will work as the main interface for the patient. The SMC app is designed for the medical staff to receive notifications from the patient room and return quick feedback to the patient.

The security layer serves as the auxiliary component of the system as it is critical whenever the patient data are moved across networks. In this application, all the data collected conform to the privacy policy with the consent of the user, and the data transmission and storage is executed in an encrypted form. For system robustness, dual systems are used in case of disruption, such as accidental break down of a server, and crash reports in the application layer and system monitoring server are used to respond to any anomalies in system behavior. The chat messages on the Internet are analyzed for author identification using machine learning algorithms [93], an intermediate security broker is used to improve data privacy and security when communication occurs between the cloud and the user [94], biometric-based authentication schemes are used to help protect patient privacy [95], an IoT security framework is used to protect smart infrastructures from cyber-attacks [96], and trusted data servers are used to protect user information using differentiated privacy protocols [97].

In summary, all the client interactions are supported by the SeVA Patient Room app (SPR) as shown in Figure 6, using an Apple built-in speech framework for both the speech-to-text and text-to-speech conversion. It uses authorization and user configuration pages for tailoring value creating services. The value fulfilling activity coordination is supported by the SeVA Master Control app (SMC) situated in a staff room where the nurses are located. It monitors the client status from the messages received regularly from the SPR, and provides the feedback from the nurses, if warranted. The SMC lets the nurses monitor the status in multiple patient rooms and process incoming requests. The nurse can send back acknowledgement, such as communicating critical movement information like “patient is sitting up” on the textbox to direct timely nurse intervention.

## 5. Conclusions 

Returning to the research question posed in Section one, in building an agile digital platform to address population health within a client ecosystem, it is necessary to consider the following two areas: (1) understanding the services needed to create value and designing digital services that leverage technologies accessible to clients to fulfill the value; and (2) designing an agile digital platform that can evolve with changing client needs by quickly reconfiguring the digital services needed to fulfill the value. With the client context changing continuously, the real-time tracking of value-in-use is needed to support the reconfiguration of digital services using both advanced technologies, such as Internet of Things (IoT), NLP, and artificial intelligence tools such as neural networks, as well as human interaction and engagement (e.g., client and nursing staff). The value of classifying service types to segment population group needs in creating value, modularization of service modules for faster reconfiguration to fulfill value, and value-in-use feedback to assess both the depth and breadth of changes in a client context to create value for the next value cycle is proposed. While no digital platform can be agile enough to support the evolving client intent, the case study uses questions as a potential strategy to surface the evolving needs of the client.

The digital platform developed to support delirium patients is currently being evaluated for its viability within a hospital setting, using a set of services considered critical to reduce patient falls. One of the goals of building an agile digital platform is to be able to add new services where there is change in the scale (e.g., new services are needed when the patient moves from the hospital to home) or scope (e.g., the condition of a patient changes and needs the engagement of caregivers to respond to needs as opposed to professional nurses in a hospital). While we can’t claim that the platform will support all future changes, both the service modularization and flexible architecture used to create and fulfill value should help its adaptation when new information sharing and activity coordination is called for. Work is on-going in supporting such agility.

## 6. Directions for Future Research

The unprecedented content from clients and partners, and the urgency to act quickly to empower and engage clients in self-managing their care within their ecosystem, is leading to the rapid digital transformation of healthcare organizations [98]. Such digital transformation, as it is linked to performance on quality, convenience, and lower costs in population health, may call for disruptive delivery models [99]. This means that administrative leadership models that have permeated healthcare organizations with a culture that values long service, organized groups of teams and professionals motivated by personal and professional standards, and external accountabilities [100], are no longer adequate. Visionary leadership and organizational change have to complement administrative leadership with enabling and adaptive leadership processes [101] to let innovative client-centric and value-focused digital services support change incrementally and a culture of learning cumulatively. A future research question is, “What agile digital services and the population groups they service can help lead to sustained digital transformation that improves performance and mitigates risks of all actors involved, including clients?”

The discussion and the case study argue for an agile digital platform that not only brings together clients and clinical and non-clinical care providers, but also a mix of human and machine actors using a set of automated and autonomous digital services dynamically to create and fulfill value. Research on the actor-network theory [102] suggests that we often fail to distinguish between the efficacy of the network and the efficacy of the processes used in the formation of the network itself. We discussed the design of digital platforms to make a network agile, where clinical and non-clinical actors used value cycle activities and digital services to address the needs of the population groups. A future research question is, “How can the digital platform expand its agility to redefine the network itself?” For example, can the digital platform used to support delirium patients learn from the client questions their evolving intent, such as the need to reduce stress or help avoid isolation by adding new digital actors (e.g., a new digital service that autonomously delivers audio or video content to suit the client’s tastes in music) or new human actors (e.g., add an automated digital service that uses teleconsultation with a mental health counselor or a social media interaction with others to exchange stories or play games)? Both of these call for altering the process used to create the network, which in turn is used to create value.

## Figures and Tables

**Figure 1 ijerph-18-05686-f001:**
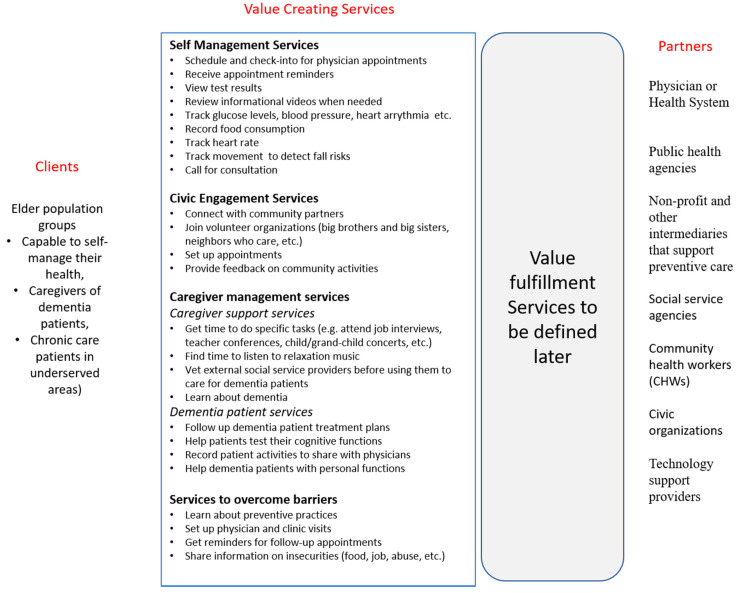
A template of services to support select population groups.

**Figure 2 ijerph-18-05686-f002:**
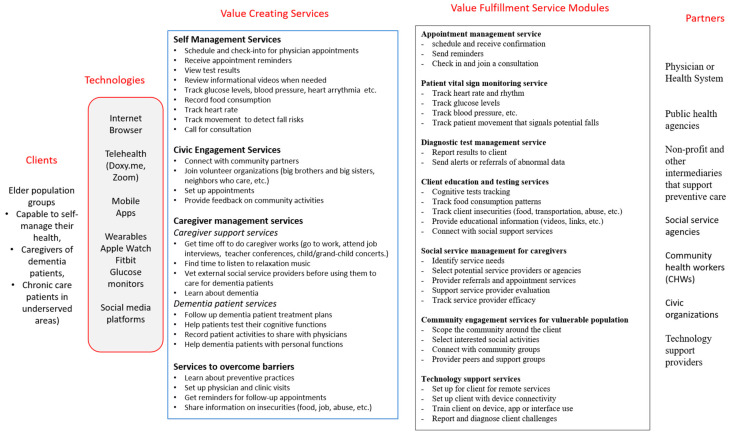
Digital platform to support digital service configuration.

**Figure 3 ijerph-18-05686-f003:**
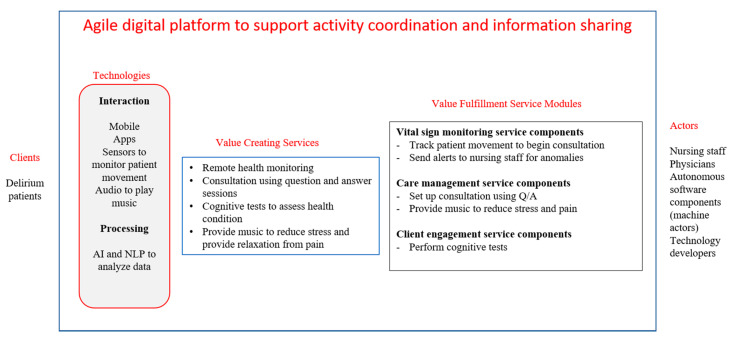
Digital platform to support delirium patients.

**Figure 4 ijerph-18-05686-f004:**
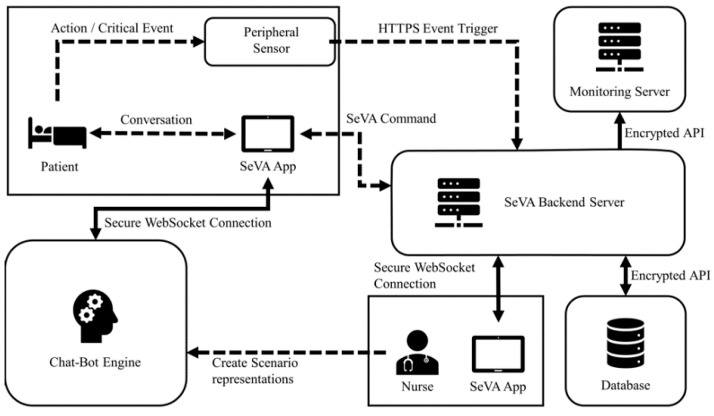
Coordination of activities among multiple actors within SeVa architecture.

**Figure 5 ijerph-18-05686-f005:**
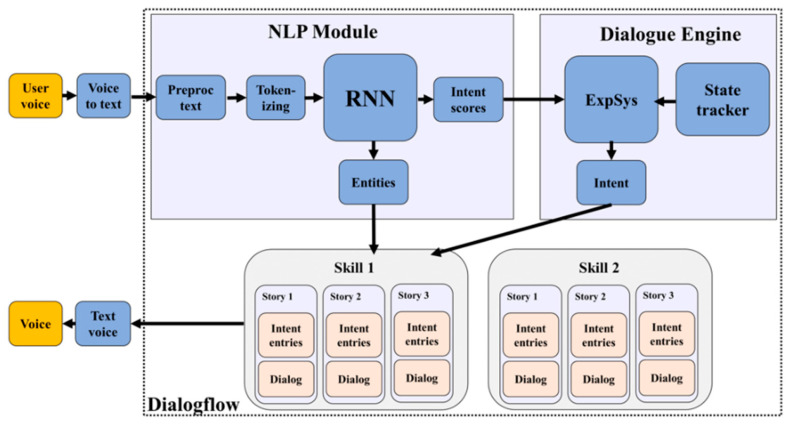
Chatbot engine skill and story design.

**Figure 6 ijerph-18-05686-f006:**
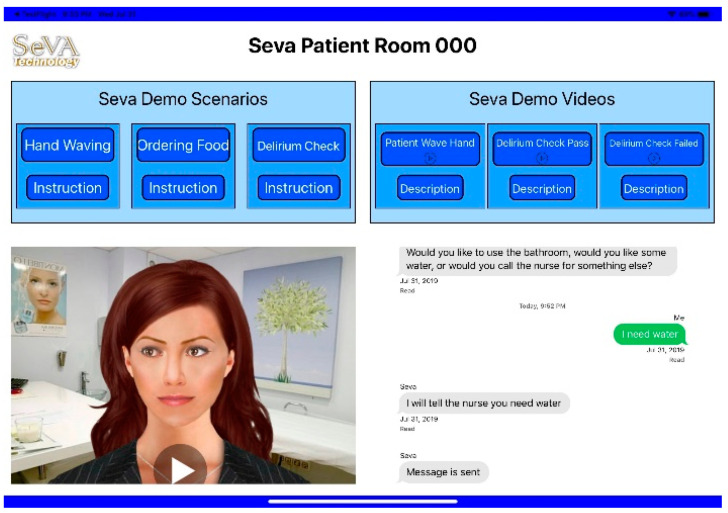
SPR application user interface.

**Table 1 ijerph-18-05686-t001:** Seva skills in the Dialogflow.

Type	Skill	Story	Description
Movement Response	Sensor Response	Fall detection	Respond to patient and notify nurse by recognizing patient movement.
		Wave	
Regular Check	Hourly Rounding	Feeling check	Perform regular hourly check to fulfill patient needs actively.
		Restroom check	
		Brace check	
		Heat pack check	
	Delirium Check	What day is today?	Perform regular delirium check to evaluate patient cognition.
		List weekdays in reversed order	
Relaxations	Soothing Music	Play music	Use music, jokes, and small talk to improve patient’s mental state.
	Small Talk	Random talk	
	Joke	Tell me a joke	
		More jokes	

## Data Availability

The authors confirm that the data supporting the digital platform development are available within the article and referenced material.

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
