# Peer review of "An Agile Digital Platform to Support Population Health—A Case Study of a Digital Platform to Support Patients with Delirium Using IoT, NLP, and AI"

_ijerph, 2021, doi:10.3390/ijerph18115686_

Round 1
Reviewer 1 Report
This is a very interesting, timely, and thought-provoking paper which describes the needs and an example system embodying a multi-component ‘agile’ system geared toward (in the example) aid and monitoring of delirium patients, a common scenario in inpatient settings. It nicely integrates many different needs, and discusses the value of different needs and components.
In the beginning paragraphs the use of jargon, repeated use of the same ‘value-‘ phrases makes it hard to read; please edit for shorter sentences and clearer language. The use of a diagram may assist. Also be aware that use of ‘value-‘ phrases may be more common to business papers than informatics ones.
Examination of the language used in informatics to describe the same things may be of use.
Apart from this there is little discussion of how such an agile system would be changed in response to new needs; one of the promises of the paper; presumably there are major benefits to an agile system over time. Discussion of flaws or possible issues in such a system is also needed. The point about difference between value in the network and in its subcomponents is important.
Minor
Line 14 – change ‘to’ to ‘should’.
Line 77 change ‘to’ to ‘in’
232 – change ‘sextually’ to ‘sexually’
Figure 1 – is something missing in the middle? If not, make the two columns more centered – at first glance the reader may assume the figure was defective – perhaps put ‘intentionally blank’ in the middle.
542- change to ‘it requires that’.
There may be minor grammatical or diction points that editors should look for.
Author Response
This is a very interesting, timely, and thought-provoking paper which describes the needs and an example system embodying a multi-component ‘agile’ system geared toward (in the example) aid and monitoring of delirium patients, a common scenario in inpatient settings. It nicely integrates many different needs, and discusses the value of different needs and components.
>> Thanks.
In the beginning paragraphs the use of jargon, repeated use of the same ‘value-‘ phrases make it hard to read; please edit for shorter sentences and clearer language. The use of a diagram may assist., Also be aware that use of ‘value-‘ phrases may be more common to business papers than informatics ones. Examination of the language used in informatics to describe the same things may be of use.
>> It is true that the value cycle concepts are frequently used in service innovation research, as this is how this paper is positioned. All sections are oriented to value creation, value fulfillment and value in use. We did clarify how the value cycle embeds the terns used in research such as informatics, used to improve its performance. This is clarified with a change in section 1 as follows:
Analyzing feedback to improve clinical diagnosis (i.e. clinical informatics), business performance (i.e. business analytics), or system performance (i.e. analytics in general) is necessary when there are changes in inputs or models used to process these inputs. When such changes are driven by evolving needs of external consumer, feedback to improve value created by an organization comes from a consumer’s perception of such value (value-in-use).
Apart from this there is little discussion of how such an agile system would be changed in response to new needs; one of the promises of the paper; presumably there are major benefits to an agile system over time. Discussion of flaws or possible issues in such a system is also needed. The point about difference between value in the network and in its subcomponents is important.
>> Agility for new needs beyond what are discussed in section 4 are identified as possible areas for future developed under conclusion. This can be viewed as a limitation. While the system currently developed is to reduce fall risk in a hospital, the structure used to develop the digital platform uses a modular approach that can be adapted if there is change in the scope or scale of the application. This is brought up in the last section.
Minor
Line 14 – change ‘to’ to ‘should’.
Line 77 change ‘to’ to ‘in’
232 – change ‘sextually’ to ‘sexually’
542- change to ‘it requires that’.
>> We have re-edited the paper to simplify the discussion.
Figure 1 – is something missing in the middle? If not, make the two columns more centered – at first glance the reader may assume the figure was defective – perhaps put ‘intentionally blank’ in the middle.
>> Figure 1 is revised to illustrate that the middle section will be coming after discussion of section 3.
Reviewer 2 Report
This paper developed a template for the design of an agile digital platform to support value cycle activities among clinical and non-clinical actors, including population groups. It illustrates design of an agile digital platform to support clients that suffer from delirium using digital services that leverage Internet of Things, natural language processing and AI that leverages real time data for learning and care adaption.
Using prior research on aging and dynamic software ecosystems, this research proposed the digital platform to support digital service configuration. A dynamic software ecosystem for rapidly evolving client expectations as shown in the digital platform needs not only increased frequency of value assessment for reconfiguration but an ability to address potential changes in scope and scale of such assessment.
On the whole, this research is readable, and the proposed digital platform is explained with extensive literature discussion and illustrated by example. Nevertheless, I thought I need more clarification from the author in this part: how the architecture proposed by the research has been proven to be feasible? If the authors can give me full description, I would like to recommend this paper to be published in Int. J. Environ. Res. Public Health.
Author Response
This paper developed a template for the design of an agile digital platform to support value cycle activities among clinical and non-clinical actors, including population groups. It illustrates design of an agile digital platform to support clients that suffer from delirium using digital services that leverage Internet of Things, natural language processing and AI that leverages real time data for learning and care adaption.
Using prior research on aging and dynamic software ecosystems, this research proposed the digital platform to support digital service configuration. A dynamic software ecosystem for rapidly evolving client expectations as shown in the digital platform needs not only increased frequency of value assessment for reconfiguration but an ability to address potential changes in scope and scale of such assessment.
On the whole, this research is readable, and the proposed digital platform is explained with extensive literature discussion and illustrated by example.
>> Thanks for your kind words.
Nevertheless, I thought I need more clarification from the author in this part: how the architecture proposed by the research has been proven to be feasible? If the authors can give me full description, I would like to recommend this paper to be published in Int. J. Environ. Res. Public Health.
>> The feasibility of the current system is currently being evaluated at a hospital after the technical viability was demonstrated. Covid19 pandemic has slowed the testing in the hospital and, upon getting approval from the Institutional Review Board, it is to be evaluated. Also, viability of this model for changes is also addressed in the conclusion section.
Reviewer 3 Report
It is interesting to see the article referencing customer centricity and the use of different types of services in common in order to measure and test the synergies between digital services and their possible evolution over time.
One of the fields where we can see better levels or a greater amount of expansion in cultural overgrowth is in the older adults, who are increasingly becoming the majority of the population pyramid.
In these times of pandemics and healthcare problems, there is an urgent need to evolve and improve those systems that, due to lack of knowledge, lack of evolution or neglect, have not been implemented and now play an important role in the health sector.
Within each country it is clear that the system differs in its objectives or perspective to be fulfilled, so it is difficult to generalise not only the data but also the results to other countries that could make use of this technology.
For some types of patients, it would be interesting to carry out a previous test or training experience in order to make a correct use of the tool and provide the user with the autonomy that it can give him/her, speeding up both public and private healthcare services.
Finally, I would like to end by saying that it seems to me to be a very interesting proposal and that it could undoubtedly be a path or line of research with great projection not only in the future but also in the present through the aforementioned shortcomings.
Author Response
>> Thanks for your support and the manuscript has been edited fully to address some of your concerns in the review. To improve clarity, we have moved some parts of section 4 to Appendix.